# Assessing Nutritional Characteristics and Bioactive Compound Distribution in Seeds, Oil, and Cake from Confectionary Sunflowers Cultivated in Serbia

**DOI:** 10.3390/foods13121882

**Published:** 2024-06-15

**Authors:** Nada Grahovac, Tanja Lužaić, Dragan Živančev, Zorica Stojanović, Ana Đurović, Ranko Romanić, Snežana Kravić, Vladimir Miklič

**Affiliations:** 1Institute of Field and Vegetable Crops, Maksima Gorkog 30, 21 000 Novi Sad, Serbia; dragan.zivancev@ifvcns.ns.ac.rs (D.Ž.); vladimir.miklic@ifvcns.ns.ac.rs (V.M.); 2Faculty of Technology Novi Sad, University of Novi Sad, Bulevar Cara Lazara 1, 21 000 Novi Sad, Serbia; tanja.luzaic@tf.uns.ac.rs (T.L.); zorica.stojanovic@uns.ac.rs (Z.S.); djurovic@uns.ac.rs (A.Đ.); rankor@uns.ac.rs (R.R.); sne@uns.ac.rs (S.K.)

**Keywords:** confectionary sunflower, cold-pressed oil, genetic variability, bioactive compounds, agricultural breeding

## Abstract

Sunflower seeds are key agricultural commodities due to their nutritional and industrial value. This study aimed to analyze the distribution of targeted bioactive compounds and assess the physical properties across 27 sunflower seed genotypes, including parental lines and F1 and F2 hybrids, cultivated in Serbia. Various analytical techniques were employed to determine the chemical composition and physical characteristics of the seeds. This research revealed significant genetic variability in fatty acid profiles, with differences in polyunsaturated and saturated fatty acid levels among the genotypes. Hybrid seeds displayed variations in 1000-seed weight and bulk density compared to parental lines, which exhibited higher essential fatty acid contents and mechanical properties advantageous for industrial processing. These insights highlight the potential for refining breeding strategies to improve seed quality for specific industrial purposes. Overall, this study emphasizes the critical role of genetic selection in enhancing the nutritional and processing qualities of sunflower seeds, offering valuable perspectives for advancing agricultural and breeding practices.

## 1. Introduction

Sunflower (*Helianthus annuus* L.) plays a crucial role in oil production, making a significant contribution to vegetable oil supplies. According to the FAO database, the global consumption of sunflower oil has demonstrated a notable increase in recent years. Sunflower oil accounts for approximately 9.5% of total vegetable oil consumption worldwide [1]. Breeding advancements have led to changes in seed characteristics, including the development of oily sunflower hybrids to increase oil content and confectionary sunflower hybrids to enhance protein content for various uses. These efforts have resulted in significant achievements, with sunflower hybrid seeds typically containing 45% to 50% oil, reaching up to 60–65%. Additionally, oily hybrids often boast a 1000-seed mass of up to 80 g and hull content below 25% [2].

Although sunflowers are primarily cultivated for oil production, confectionary hybrids also are of great importance. These hybrids yield larger seeds with distinctive colors and shapes, thicker hulls, and higher hull content, often ranging from 40 to 50% [3]. Additionally, the 1000-seed mass typically exceeds 100 g [4]. These hybrids require specific traits, including increased protein content exceeding 25%, enhanced essential amino acids, reduced oil content to below 40%, enhanced oil stability, uniform seed characteristics, and improved seed dehulling properties [2]. Seeds from these hybrids are relished as snacks in their raw or baked form and serve as integral ingredients in various food products, including bakery items [5], halva [6], and sunflower butter [7].

Sunflower seeds, the predominant output of sunflower cultivation, are particularly rich in bioactive compounds, including vitamins, minerals, antioxidants, and phytochemicals, which are known for their potential health benefits [8,9].

Sunflower seed oil is highly sought after worldwide, especially in Europe, due to its wide-spread availability and numerous health benefits, including cholesterol reduction, antioxidant properties, and blood pressure regulation [10,11]. It is noteworthy that sunflower oil boasts a high content of linoleic acid (C18:2, omega 6) ranging from 48% to 74%, making it a valuable source of this essential fatty acid [12]. In contrast to soybean and canola oil, sunflower oil contains minimal levels of saturated fatty acids such as palmitic and stearic acid, and it lacks significant amounts of α-linolenic acid (ALA, omega 3). Sunflower oil contains minor components, constituting approximately 0.5–1.5% of its composition, including phospholipids, free fatty acids, tocopherols, pigments, alcohols, phytosterols, carotenoids, and phenolic acids. Tocopherols and phytosterols contribute to both health benefits and oxidative stability [13], while other molecules can induce oxidative rancidity and contribute to off-flavor development during processing and storage. During the process of oil extraction from sunflower seeds, cake generated as a byproduct is a valuable source of protein and bioactive compounds [9,14,15], improving its nutritional quality. Sunflower seed cake, owing to its significant antioxidant content, has potential for diverse technological applications, as highlighted by Weisz et al. [16], underscoring sunflower by-products as substantial sources of natural antioxidants and phenolic components. Moreover, they are reservoirs of bioactive phytochemicals, including phenolic compounds, and other molecules renowned for their antioxidant, antidiabetic, antihypertensive, and chemopreventive properties. In addition, the chemical composition of oil processing remnants varies based on factors such as seed origin, the method of oil extraction, and its efficacy [17]. Due to its inherent chemical composition, a significant portion of this waste can be utilized in valuable ways, such as in animal feed production, direct incorporation in food preparation, or the extraction of its bioactive components.

Sunflower seed varieties, despite their abundance, are often harvested without scientific classification, leading to mixed outcomes and lower product quality and efficiency. This limitation hinders the high-quality development of the sunflower seed processing industry. Recent advancements in sunflower seed research have primarily centered on areas such as the quality of seeds [18], oil [19], and protein [8]. However, further research is necessary to explore the raw material properties, physicochemical characteristics, and processing attributes of sunflower seeds, as available reports on these aspects are limited. Their industrial applications often require the screening or grading of raw materials, emphasizing that parameters such as the dimensions, sphericity, weight, hardness, and fatty acid composition variation in sunflower oil among different varieties warrants more investigation. Seed characteristics significantly influence the design and operation of agricultural machinery for drying, dehulling, storage, and oil extraction, with seed size crucial for harvester design and adjustment [20,21,22]. However, there is currently a knowledge gap regarding the interplay between the geometric and physical–mechanical properties of seeds and their nutritional attributes, encompassing seeds, oil, and cake, within both parental lines and the subsequently developed F1 and F2 hybrids of confectionary sunflower. Additionally, research on the phytochemical composition of the parent lines and the resulting F1 and F2 hybrid genotypes in confectionary sunflowers remains limited. Moreover, the process of screening phytochemicals within the extensive populations utilized in breeding programs proves to be challenging, time-intensive, and costly.

This research explores the comprehensive nutritional composition of confectionary sunflower seeds, oil, and cake, encompassing both parental lines and their subsequent F1 and F2 hybrids. Our findings aim to provide valuable insights for food manufacturers in industry by thoroughly examining the interrelations between these nutritional constituents and the geometric and physical–mechanical characteristics of seed parameters. By applying these insights, manufacturers can enhance the nutritional profiles and antioxidant capacities of their final products, thereby contributing to the advancement of healthier and more delicious food offerings.

## 2. Materials and Methods

### 2.1. Materials, Sample Preparation, and Reagents

An extensive examination was carried out on 27 genotypes of confectionary sunflower seeds. The investigation encompassed restorer lines (RL1 and RL2), cytoplasmic male sterile paternal lines (CMS1–CMS5), and experimental hybrids of the F1 generation (FG1–FG10) (F1 genotypes, hybrid seeds), as well as the F2 generation (SG1–SG10) (F2 genotypes, commercial seeds) of confectionary sunflower. Agricultural producers procure hybrid seeds for sunflower cultivation to generate the F2 generation (commercial seeds), which, upon harvest, have diverse applications across industries and food sectors. These hybrids were cultivated through conventional selection techniques, yielding two-line hybrids by crossing restorer lines with cytoplasmic male sterile lines, incorporating fertility restoration genes. These samples were grown at the experimental fields of the Institute of Field and Vegetable Crops in Novi Sad (between 45°460′2700″ N, 19°60′4400″ E and 44°520′1400″ N, 20°380′2500″ E m above sea level), Serbia, as part of the Institute’s breeding program. Hybrids and lines were cultivated under standard conditions, without irrigation, following a randomized complete-block design with three replications. Restorer and CMS paternal lines were mechanically sown at a spacing of 70 × 24 cm^2^, followed by manual harvesting. Experimental hybrids of the F1 generation (FG1–FG10) were sown with a row-to-row spacing of 70 cm, and CMS lines within a row were spaced 25–26 cm apart, while restorer lines were spaced 35–36 cm apart within a row. The sowing arrangement comprised four rows of CMS lines and two rows of restorer lines, followed by four rows of CMS lines, an interruption of two rows, another four rows of CMS lines, two rows of restorer lines, and finally, four rows of CMS lines. Sowing involved two rows of restorer paternal lines and eight rows of CMS lines. Two unsown rows facilitated ease of passage during flowering monitoring and the removal of atypical plants. Machine harvesting was conducted subsequently. Experimental hybrids of the F2 generation were sown within a plot measuring 10.5 m^2^ (0.7 × 0.25 × 5 m^3^), arranged in four rows with three repetitions. Machine harvesting was performed, extracting seeds from the two central rows while excluding edge plants.

Subsequently, seeds from each line and hybrid were pooled, thoroughly mixed, and standardized into three representative replicates, each comprising 5 kg for oil mechanical extraction and 1 kg for seed analysis. These samples were stored at 20 °C until pressing and analysis. Each hybrid’s and line’s seeds, replicated three times, underwent cold-pressing using a screw press with a designed capacity of 25–30 kg h^−1^ and electric motor power of 2.2 kW, at a frequency of 33–34 min^−1^. The press, manufactured by Mikron SZR, Temerin, Serbia, is commonly used for cold-pressed oil production in Serbia. Following pressing, the output yielded two products: oil and cake. The temperature of the extracted sunflower oil ranged from 55 to 60 °C for all investigated samples. Cakes from each pressing were stored at 20 °C for further testing, while the extracted oil underwent further analysis.

The chemicals and reagents utilized were of analytical reagent quality.

### 2.2. Assessment of Raw Material Quality

#### 2.2.1. Basic Chemical Composition of Sunflower Seed and Cake

The moisture content (MC) in the samples of confectionery sunflower seed and cake was analyzed according to the ISO 665, 2000 standard [23]. Oil content (OC) was determined using the ISO 659, 2009 standard [24] method, utilizing n-hexane as the solvent in Soxhlet apparatus (Soxtherm 2000, Gerhardt, Bonn, Germany). Protein content (PC) in the samples of confectionery sunflower seed and cake was analyzed according to the American Association of Cereal Chemists (AACC) methods 46-16.01 [25]. The results for moisture, oil, and protein content were reported in grams per 100 g of confectionery seeds.

#### 2.2.2. Engineering Characteristics of Sunflower Seeds

##### Investigation of Seed Dimensions

The length (L), width (W), and thickness (T) of the seeds were measured using a vernier caliper with an accuracy of ±0.05 mm. A sample of 100 randomly selected sunflower seeds was analyzed to determine their dimensions.

##### General Characteristics of Seed

The 1000-seed mass (Mts) was calculated by weighing 250 randomly selected sunflower seeds with 0.001 g precision and extrapolating the result to 1000 seeds, based on dry weight. Three consecutive measurements were averaged for accuracy. Hull content (Hc) was assessed by manually removing hulls from 10 g of seeds, with the result presented as a percentage of the total mass. Based on the hull content, as the remaining portion of the sunflower seed is the kernel, the hull/kernel ratio (h/k) was also calculated.

##### Gravimetric Characteristics of Seeds

True density (Td), calculated as the ratio of sample mass to true volume, was determined using a precision electronic balance (0.001 g) and a calibrated 100 mL volumetric flask (using the 60% ethanol displacement method). Bulk density (Bd), representing the ratio of seed mass to total volume, was measured with standard equipment (250 mL total volume with air-displacement piston). Each test was conducted thrice independently. Porosity (P), indicating the space within bulk seeds not occupied by seeds, was computed using Equation (1):(1)P=(1 − (Bd/Td)) × 100

##### Seed Geometry Analysis

Equivalent diameter (dp) in millimeters was calculated using Equation (2):(2)$${\mathrm{dp}=(L\;\times \;W\;\times \;T)}^{1/3}$$

Sphericity (Sp) was derived from Equation (3):(3)$${\mathrm{Sp}=\mathrm{dp}\;\times \;L}^{-1}$$

Surface area (S) was computed based on equations provided by McCabe et al. [26] as well as Malik and Saini [27] (Equation (4)):(4)$${S=\pi \;\times \;\mathrm{dp}}^{2}$$

The calculation of seed volume (V) was based on Equation (5):(5)$${V=(\pi \;\times \;\mathrm{dp}}^{3})/3$$

### 2.3. Examining the Fatty Acid Profile and Fatty Acid Ratio of Sunflower Seed Oil

The fatty acid profiles of cold-pressed sunflower oils were analyzed via gas chromatography–mass spectrometry (GC-MS) following ISO methods 12966-4:2015 [28] and 12966-2:2017 [29]. An Agilent Technologies gas chromatograph coupled with a mass-selective detector was utilized, equipped with a DB-23 Agilent Technologies capillary column. The GC conditions included a temperature range from 50 °C to 230 °C, with helium as the carrier gas. Fatty acids were identified using retention times and mass spectra databases, with a Supelco standard (37 FAMEs) solution used for peak identification [30]. Results were expressed as percentages of total fatty acids. The fatty ratio of oleic acid to linoleic acid (O/L) was determined by dividing the percentage of oleic acid by the percentage of linoleic acid.

### 2.4. Analysis of the Bioactive Components of Sunflower Seed, Oil, and Cake

Total tocopherol content (TTs) was assessed via high-performance liquid chromatography (HPLC) following the ISO 9936:2016 standards [31]. Total tocopherols were extracted from sunflower seeds and oil under a nitrogen atmosphere using 99% ethanol in the presence of potassium hydroxide. The total phenolic content (TPc) was determined from cake using the Folin–Ciocalteu spectrophotometric method [32,33,34] with slight modifications. Sunflower seed extract (0.5 mL) was mixed with doubly distilled water (6.0 mL) and Folin–Ciocalteu’s reagent (0.5 mL) in a 10 mL volumetric flask for 60 s. Then, 15% sodium carbonate solution (2.0 mL) was added, followed by incubation in a dark place for 45 min. Absorbance was measured at 760 nm using a UV–Vis spectrophotometer. Quantification utilized a standard calibration curve with chlorogenic acid, and the total phenolic content was expressed as milligrams of chlorogenic acid (CGA) equivalent per kilogram of sample dry weight (mg CGA eq kg^−1^). The total carotenoid content (TC) of cold-pressed sunflower oils was estimated by absorbance measurement at 445 nm, following the procedure described by Djalović et al. [34], and expressed as mg of β-carotene per kg of oil (mg β-C eq kg^−1^). The chlorophyll content of cold-pressed sunflower oils, represented as pheophytin a (mg Phe A eq kg^−1^), was determined by measuring oil absorbance at 667 nm according to the method described by Pokorny et al. [35].

### 2.5. Pressing

The oil extraction efficiency (E) was determined by contrasting the oil content in the press cake with the initial oil content in the seeds, as per Equation (6) [36]:(6)$$E=100\cdot \left[1-\frac{\frac{{\mathrm{OC}}_{s}}{\left(100-{\mathrm{OC}}_{s}\right)}}{\frac{\mathrm{OC}}{\left(100-\mathrm{OC}\right)}}\right]$$

E represents the efficiency of oil extraction from sunflower seeds (measured as a percentage), OC indicates the initial oil content in the seed (expressed as a percentage, i.e., grams of oil per 100 g of seed), and OCs denotes the oil content in the press cake (measured as a percentage, i.e., grams of oil per 100 g of seed).

### 2.6. Statistical Analysis

The data underwent analysis through one-way analysis of variance (ANOVA), supplemented by Tukey’s Honestly Significant Difference (HSD) test, to discern notable differences at a significance level of *p* < 0.05. Statistical analysis and Artificial Neural Network (ANN) modeling were conducted using Statistica version 13.5.0.17 (StatSoft, Tulsa, OK, USA). Principal Component Analysis (PCA), linear regression analysis, and Pearson correlation matrix analysis were performed using XLSTAT BASIC, version 5.1, 2022 software (Addinsoft, Paris, France).

## 3. Results and Discussion

### 3.1. Basic Chemical Composition of Seeds

This study aimed to analyze the distribution of the oil, protein, and moisture levels within the seeds of the parental lines, as well as the F1 and F2 generations, of sunflowers produced through conventional crossing methods (Table 1).

Notable variations in basic chemical composition were observed among the tested samples. Specifically, within the parental group, the CMS4 line of cytoplasmic female sterile paternal lines exhibited the statistically highest oil content of all parental lines at 35.82%. Additionally, hybrid CMS4 not only has a high oil content but also boasts the highest total tocopherol content at 254.51 mg kg^−1^ in the seed. In the same group, the CMS5 line displayed the statistically highest protein content of all parental lines at 22.70%, while the RL2 parent line had the statistically lowest protein content of all investigated samples at 13.13%. Transitioning to the hybrid generations, the FG7 hybrid in the F1 generation showcased the highest oil content at 28.84%, which was not statistically different from the FG3 and FG8 hybrids from the F1 generation and the SG7 hybrid from the F2 generation, whereas the FG5 hybrid in the F1 generation demonstrated the statistically lowest content at 19.35%. Conversely, the FG9 hybrid in the F1 generation exhibited the highest protein content at 22.03%, which was not statistically different from the FG6 hybrid in the F1 generation, while the SG1 hybrid in the F2 generation displayed the lowest content at 13.43%, which was not statistically different from the SG2 hybrid in the F2 generation. Additionally, aside from its high oil content, hybrid FG9 also boasted the highest total tocopherol content at 173.12 mg kg^−1^ in the seeds. Conversely, hybrid FG1 within the F1 generation showed the lowest total tocopherol content at 95.24 mg kg^−1^.

This illustrates how variability in the oil, protein, and total tocopherols of hybrid seeds arises from crossing parental lines. Recent research has highlighted the potential of sunflower proteins as valuable components for human nutrition and biofilm production [37,38]. Studies have consistently demonstrated the safety and nutritional benefits of sunflower protein products for human consumption [39,40]. These findings suggest that the resulting hybrids can serve various purposes in the food industry, depending on specific needs. In terms of technology, tocopherols play a crucial role in safeguarding oil against oxidative degradation, particularly by targeting the double bonds present in unsaturated fatty acids [41]. The total tocopherol content in the seeds (95.24–173.12 mg kg^−1^) observed in our study aligns with the results reported by other researchers. Velasco et al. [42] and Žilić et al. [43] similarly reported the content of tocopherols in sunflower seeds, with ranges of 119 to 491 mg kg^−1^ and 200.67 to 220.05 mg kg^−1^. The variations in the total tocopherol content in seeds can be attributed to differences in genotypes, cultivation methods, as well as sampling and analytical techniques.

Regarding water content, the parental lines (ranging from 5.66% to 7.04%) closely resembled the F1 and F2 generations (ranging from 5.41% to 6.57%). This similarity is noteworthy as water content can act as a medium for the Maillard reaction during roasting, influencing the aroma and flavor of roasted sunflower seeds [44]. Moreover, water content levels impact storage life and freshness, necessitating caution to prevent moisture-related spoilage and mold formation during storage. It is essential to maintain proper storage conditions for both parent lines and their resulting hybrids to mitigate or completely avoid such risks [45].

### 3.2. Sunflower Seed Engineering Characteristics

The size attributes and quality of sunflower seeds play a crucial role in the harvest, cleaning, and grading processes within the entire sunflower seed processing chain [46]. Table 2 and Table 3 present the dimensional, geometric, technical, and technological characteristics of seeds from the parental lines, as well as the F1 and F2 generations.

The dimensions of the investigated sunflower seeds ranged from 11.33 to 20.68 mm in length, from 4.53 to 9.04 mm in width, and from 2.89 to 6.12 mm in thickness. Sunflower seeds from the F1 and F2 generations showed slightly larger lengths (11.33 to 20.68 mm) compared to the parental lines (12.58–19.23 mm), with no notable differences in width, thickness, or equivalent diameter (dp) observed among the tested genotypes. This suggests that sorting based on seed length during the cleaning and grading phases could be a viable evaluation method. The size of the parental sunflower seeds was generally smaller compared to the sunflower hybrid seeds, except in the case of length, where that of CMS1 was not statistically diverse from that of the FG2, SG1, and SG2 hybrids, and the width and thickness of CMS3 were was statistically higher than the width and thickness of the other examined sunflowers samples. Minor surface area variations were detected among the parental lines (101.52–246.80 mm^2^) and the F1 and F2 generations (129.01–214.63 mm^2^), with similarities noted in seed volume. The father lines possessed the smallest surface area of all the examined samples. The sphericity of the sunflower seeds varied from 0.40 to 0.72.

These parameters are vital for designing seed hoppers optimized for sunflower seed quality screening and classification. In the hybrid sunflower seeds (F1 and F2 generations), the 1000-seed weights (Mts) were higher (69.10 to 139.63 g) compared to those of the father lines (53.09 to 53.62 g), while in the case of bulk density (Bd), no differences were observed. However, the parental lines had a higher hull/kernel ratio (h/k) and true density (Td) (0.42 to 0.92 and 460.64 to 711.86 kg m^−3^, respectively) compared to the hybrid seeds (0.39 to 0.88 and 509.11 to 704.93 kg m^−3^). No differences were observed in porosity and hull content (Hc) among the tested samples. The comparison between parental lines and hybrid sunflower seeds underscores the significance of size and quality attributes in seed processing. While the parental lines exhibited certain advantages in metrics like hull/kernel ratio and true density, the hybrids demonstrated superiority in aspects like 1000-seed weight and bulk density. These insights inform tailored approaches for optimizing equipment and processes within the food industry, ensuring the efficient screening and classification of sunflower seeds. Comparing these results with those of other researchers, similar trends have been observed [4,20,47]. Additionally, Li et al. [45] observed slight variations in the sphericity of edible sunflower seeds, underscoring the significance of this characteristic in the mechanical processing of seeds. Seeds with sphericity values close to 1 exhibit a higher tendency to rotate around any of their three principal axes. This property is essential for the effective design of seed hoppers, as it influences the flow and handling of seeds during processing. Moreover, Petraru et al. [46] found that sunflower seeds generally exhibit superior physical attributes such as seed weight and size, which are critical for processing efficiency.

### 3.3. Examining the Fatty Acid Profile of Sunflower Seed Oil

The nutritional significance of sunflower oil, particularly its fatty acid composition, holds paramount importance in human nutrition and industrial applications. The fatty acid composition significantly influences both the nutritional and technological properties of the oil [48]. Sunflower seeds are rich in various fatty acids, including linoleic acid (C18:2), oleic acid (C18:1), stearic acid (C18:0), palmitic acid (C16:0), among others [49]. The fatty acid composition of the tested genotypes is shown in Figure 1A.

In general, oils derived from parental lines exhibited high levels of polyunsaturated fatty acids (PUFA) and low levels of saturated fatty acids (SFA), ranging from 41.65% to 64.63% and 11.60% to 16.15%, respectively. The obtained hybrids (F1 and F2 generations) displayed slightly lower values, with PUFA ranging from 41.49% to 57.86% and SFA from 10.63% to 14.27% (Figure 1B). This composition is deemed favorable from a health perspective. Linoleic acid (C18:2) was the predominant polyunsaturated fatty acid, constituting the major component, ranging from 41.65% to 64.63% in the parental lines and from 41.49% to 57.86% in the F1 and F2 hybrids. Among the monounsaturated fatty acids, oleic acid (C18:1) dominated, with content ranging from 22.24% to 45.17% in the parental lines and from 30.04% to 44.39% in the hybrids. Regarding saturated fatty acids, palmitic acid (C16:0) content ranged from 5.60% to 7.4% in the parental lines and from 5.42% to 7.01% in the hybrids, while stearic acid (C18:0) ranged from 3.43% to 7.83% in the parental lines and from 3.54% to 6.02% in the hybrids. Oleic acid and saturated fatty acids are technologically significant due to their beneficial effects on oil stability. Both oleic acid and SFAs enhance the oxidative and thermal stability of oils, which improves the longevity and quality of edible oil products [50]. The content of myristic, stearic, γ-linolenic, arachidic, eicosenoic, behenic, and lignoceric acids remained below 2% for both examined groups (Figure 1A). Among the tested genotypes, the oils from parental lines RL1, CMS3, RL2, and CMS2 exhibited the highest levels of linoleic (64.63%), oleic (45.2%), palmitic (7.4%), and stearic (7.8%) acids, respectively. Further, in hybrids SG1 and FG6, the highest content of linoleic (57.86%) and palmitic (7.01%) acids was recorded. Hybrid FG3 had the highest content of oleic (44.4%) and stearic (6.02%) acids and the lowest content of linoleic (41.49%) acid. In contrast, the lines CMS4, RL1, CMS3, and CMS5 displayed the lowest contents of linoleic (41.65%), oleic (22.24%), palmitic (5.60%), and stearic (3.43%) acids, respectively. In addition, the lowest contents of palmitic (5.42%), stearic (3.54%), and oleic (30.04%) acids were measured for hybrids SG5, SG6, and SG2, respectively. The findings reported by other researchers corroborate our results [45,51,52].

Oleic acid (C18:1) and linoleic acid (C18:2) are essential fatty acids crucial for human health. Research has verified their positive effects on blood lipids [53,54], anti-inflammatory responses [55], and combating atherosclerosis [56]. Additionally, the ratio of oleic acid to linoleic acid (O/L) in oil holds paramount importance for its application. According to Wei et al. [57], a ratio of 1:1 is suitable for frying oil. Among all tested genotypes, which encompass parental lines and hybrids, the ratio of oleic acid to linoleic acid in diverse varieties of sunflower seed oil ranged from 1:0.34 to 1:1.07 (Figure 1), highlighting the substantial genetic potential for enhancement. The substantial presence of vital linoleic acid (41.50–64.63%) further enriches the nutritional profile of sunflower seed oils (hybrid seeds) for human consumption. Notably, CMS3, CMS4 (parent lines), and FG3 (hybrid) exhibited a ratio close to 1:1, making them suitable candidates for specialized frying oil processing based on their fatty acid composition. The notable differences in the fatty acid composition of sunflower seeds may stem from varietal disparities, particularly in high-oleic-acid, high-linoleic-acid, and high-stearic-acid varieties, which significantly influence the overall fatty acid profile [45].

### 3.4. Analysis of the Bioactive Components of Sunflower Seed Oil and Cake

The examination of 27 genotypes, comprising seven parental lines and 20 F1 and F2 genotypes, provided valuable insights into their nutritional composition and antioxidant profiles, as detailed in Table 4. These findings are pivotal for evaluating the potential health advantages associated with these genotypes and their applicability across various agricultural and dietary contexts.

This analysis encompassed the measurement of bioactive compound levels to assess the distribution of antioxidants within both oil and cake, alongside evaluating moisture, oil, and protein content in the cake. Given the diverse chemical properties, solubility, and polarity of the compounds under scrutiny, it is anticipated that some bioactive compounds may not be coextracted with the oil. Notably, water-soluble bioactive compounds predominantly remained in the cake.

Carotenoids and chlorophylls, natural pigments found in unrefined oils, not only influence color but also impact oil stability and shelf life due to their antioxidant properties [35,58]. The total carotenoid content (TC) ranged from 3.06 to 14.19 mg/kg in the examined oil samples. Its highest content in all examined parental lines was in the mother line CMS3 (14.19 mg kg^−1^), which was statistically higher than that in the F1 and F2 generations (FG1, FG3, FG4, FG7–FG10, and SG2–SG10). In the case of total chlorophyll content (TCl), we noticed the opposite trend. Namely, this parental line possessed the statistically lowest content (TCl) (0.00 mg kg^−1^) of all the parental lines. Also, it was statistically lower than that in the F1 and F2 generations (FG1–FG4, FG6, FG8–FG10, SG1, SG4–SG6, SG9, and SG10), a characteristic of unrefined sunflower oil. Tuberoso et al. [59] reported a total chlorophyll content of 2.3 mg kg^−1^ in sunflower oil. The TC and TCl in cold-pressed non-oil sunflower seeds can vary depending on several factors, such as the variety of sunflower, growing conditions, and processing methods.

The analysis revealed no notable variations in PCc and MCc among the parental lines and the resulting F1 and F2 genotypes of sunflower cake, as indicated in Table 3. However, significant discrepancies were observed across the genotypes concerning OCs (ranging from 17.04 to 44.38% and 17.49 to 26.30%, respectively) and TPc levels (ranging from 602.27 to 780.66 mg CGA eq kg^−1^ and 516.12 to 810.83 mg CGA eq kg^−1^, respectively). The CMS4 parental line exhibited the significantly highest total OCs at 44.38%, while the FG8 genotype from the F1 generation showed a content of 26.30%, which was the highest of all examined F1 and F2 genotypes. In contrast, the RL1 parental line and SG2 genotype from the F2 generation displayed similar OCs in the cake, with values of 18.08% and 18.47%, respectively. The SG10 genotype from the F2 generation exhibited the highest TPc, reaching 810.83 mg CGA eq kg^−1^. Conversely, the FG10 hybrid from the F1 generation displayed the lowest TPc at 516.12 mg CGA eq kg^−1^. This variation in polyphenol content implies that certain genotypes may provide greater health benefits due to heightened antioxidant capacity. In contrast, Weisz et al. [16] noted elevated total polyphenol levels in their examined non-oilseed sunflower kernel samples, ranging from 3291.9 to 3611.0 mg per 100 g of dry matter. These variations in total polyphenol content can be attributed to differences in genotypes, cultivation methods, and harvesting practices. Additionally, variations may arise from differences in sampling, sample preparation, and the analytical techniques employed during analysis.

### 3.5. Principal Component Analysis (PCA)

To explore the influence of fundamental chemical composition, grain physical properties and fatty acid, chlorophyll, and tocopherol contents on the heritage of various sunflower hybrids, the data underwent comprehensive compositional PCA analysis. The first principal component (PC1) captured over 30% of the variability, while the second principal component (PC2) elucidated over 25% of the variability (Figure 2).

Palmitic acid (C16:0) content exhibited a positive correlation with Td, Bd, and total chlorophyll content (TCl), while displaying a negative correlation with lignoceric acid (C24:0), V, S, dP, Mts, T, and W. Total tocopherol content (TTs), protein content (PC), and oil content (OC) showed positive correlations with three forms of saturated fatty acids (stearic C18:0, arachidic C20:0, and behenic C22:0) and one form of unsaturated fatty acid oleic acid (C18:1), but displayed negative correlations with unsaturated fatty acids (linoleic C18:2 and eicosenoic C20:1), as well as specific grain physical properties, including h/k, P, Hc, L and moisture content (MC).

Restorer lines and three sunflower hybrids (FG1, FG2, and FG6) demonstrated close associations with C16:0, Td, Bd, and TCl. Conversely, a mother line (CMS3) and four sunflower hybrids (FG5, FG9, SG6, and SG8) exhibited strong connections with C24:0, V, S, dP, Mts, T, W, and Sp. Three mother lines (CMS2, CMS4, and CMS5) and five sunflower hybrids (FG3, FG7, FG8, FG10, and SG4) were closely linked to TTs, PC, OC, C18:0, C20:0, C22:0, and C18:1. Additionally, according to the PCA biplot, one mother line (CMS3) and sunflower hybrids with high PUFA contents (SG1, SG2, SG3, SG5, SG7, and SG10) were identified.

To comprehensively understand how fundamental processing parameters influence the inheritance of different sunflower hybrids, we subjected the results to a robust compositional PCA. The first principal component (PC1) explained over 40% of the variability, while the second principal component (PC2) elucidated around 35% of the variability (Figure 3). TPc demonstrated a negative correlation with protein content. Oil extraction efficiency (E) showed a negative correlation with the oil content in cake (OCs). One father line (RL2) and two mother lines (CMS1, and CMS4), along with five sunflower hybrids (FG1, FG2, FG3, FG7, and SG8), were closely associated with E. Conversely, three sunflower hybrids (SG5, SG6, and SG8) along with one mother line (CMS5) were closely linked to the percentage of oil in cakes (OCs). One mother line (CMS2) and five sunflower hybrids (FG4, FG5, FG6, FG9, and FG10) were closely associated with the protein content in cake. The remaining nine examined sunflower samples exhibited high TPc in cake.

Pearson’s correlation coefficients were determined to find relationships among the examined nutritional composition, physical properties of sunflower seeds, and processing parameters (Table 5). Firstly, regarding the physical properties of sunflower seeds, S showed a statistically significant positive correlation with dP and V (r = 0.99, 0.99), and dP and V (r = 0.99), as presented in Table 5. Also, Mts was significantly positively correlated with the three above-mentioned physical properties of sunflower seeds, dP, S, and V (r = 0.80, 0.78, and 0.76, respectively). Conversely, significant negative correlations were observed between specific weight (Td) and the previously mentioned physical properties of sunflower seeds. Additionally, a significant negative correlation of −0.53 was found between porosity and Sp, whereas significant positive correlations were found between porosity and h/k and Td (0.48 and 0.49, respectively). Moreover, a statistically significant negative correlation was established between some of the examined chemical parameters of sunflower seeds, total tocopherol content (TTs), protein content (PC), and oil content, (OC) and two physical parameters, porosity and h/k. Significant positive correlations were also found between the contents of behenic (C22:0) and lignoceric acids (C24:0) and certain physical properties of sunflowers seed (S, dP, and V), whereas the content of palmitic acid (C16:0) exhibited negative correlations with the above-mentioned physical properties of sunflower seeds. Furthermore, significant positive correlations were observed between unsaturated fatty acid (linoleic C18:2) and Td (0.45), whereas oleic acid (C18:1) showed significant negative correlations with this parameter (−0.46). Conversely, saturated fatty acid (behenic acid, C22:0) exhibited significant negative correlations with Td (−0.66).

Similarly the chemical–nutritional parameters PC and TT displayed significant positive correlations (0.52, 0.54, 0.53, and 0.58, respectively) with oleic acid (C18:1) and behenic acid (C22:0), whereas linoleic acid (C18:2) showed significant negative correlations (−0.54 and −0.52) with them. As expected, E was statistically significantly positively correlated with oil content, while a positive correlation was found between the percentage of oil in the cake and porosity. However, noteworthy negative correlations were observed, whereby Td and P were negatively correlated with OCs (−0.48 and −0.50, respectively), and h/k and P with PCc (−0.46 and −0.48, respectively).

## 4. Conclusions

This study investigated the chemical composition and physical properties of sunflower seeds from parental lines and their F1 and F2 generations derived through conventional crossing methods. Significant disparities were found, with notable examples including the CMS4 line showing the highest oil content (35.82%) and the CMS5 line displaying the highest protein content (22.70%). Among the hybrids, FG7 in the F1 generation had the highest oil content (28.84%), while FG5 had the lowest (19.35%). Similarly, the protein content was highest in FG9 from the F1 generation (22.03%), and the lowest in SG1 from the F2 generation (13.43%). These findings illuminate the complex dynamics of hybrid seed diversity and their implications for food industry applications. Remarkably, hybrid FG9 exhibited the highest total tocopherol content (173.12 mg kg^−1^), whereas FG1 within the F1 generation showed the lowest (95.24 mg kg^−1^), underscoring the influence of parental lineage crossing on seed variability and nutritional implications. A comparison between parental lines and hybrid seeds revealed differences in key quality attributes such us 1000-seed weight and bulk density, with hybrids showing superior performance. Parental lines, however, excelled in aspects like hull/kernel ratio and true density, important for optimizing food industry equipment and processes. This study also highlighted substantial variations in fatty acid composition, particularly oleic and linoleic acids, emphasizing the genetic potential for enhancing the nutritional profile of sunflower seed oils. Varied levels of bioactive compounds, notably polyphenols, suggest differences in antioxidant capacity among genotypes. This variation emphases the potential for genetic enhancement to optimize the health benefits of sunflower seed oils. Moreover, this study underscores the influence of various factors on sunflower seed oil and cake composition, with cold extraction emerging as the preferred method for producing high-quality oils. In conclusion, the distinct attributes between parental lines and their resultant hybrids underscore the necessity for comprehensive quality assessments. Such evaluations are essential to ensure the selection of genotypes that meet specific nutritional and processing standards, considering their utilization as raw materials within the food industry.

## Figures and Tables

**Figure 1 foods-13-01882-f001:**
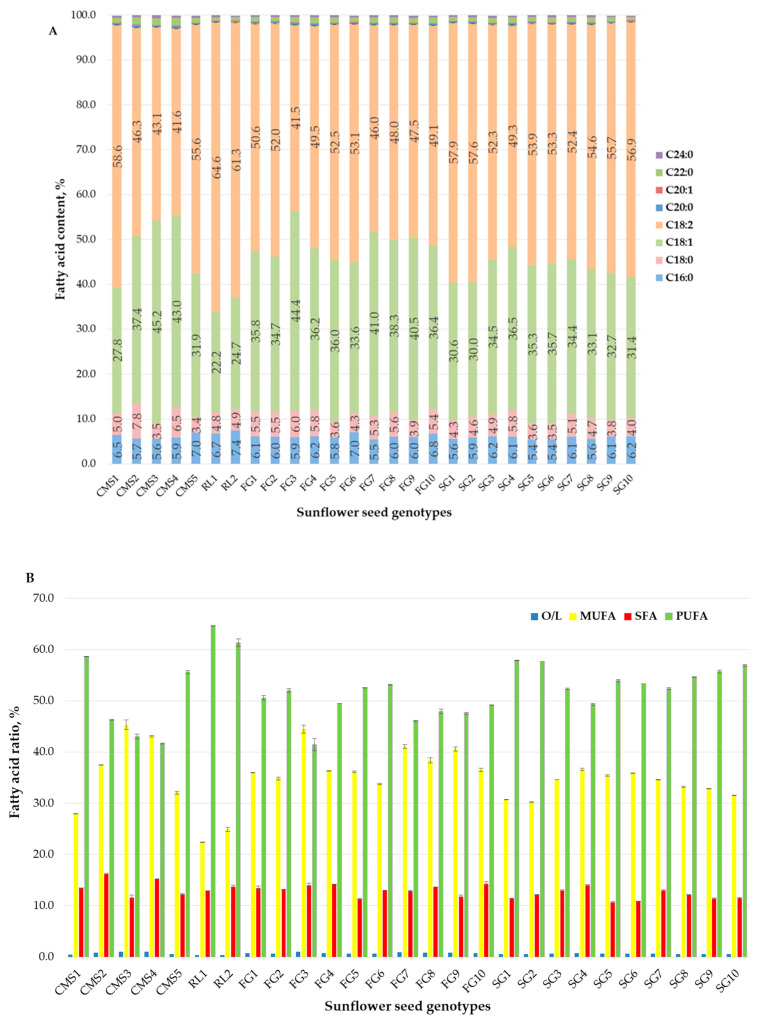
Fatty acid composition of sunflower seed oil with different genotypes: (**A**) fatty acid composition; (**B**) fatty acid ratio.

**Figure 2 foods-13-01882-f002:**
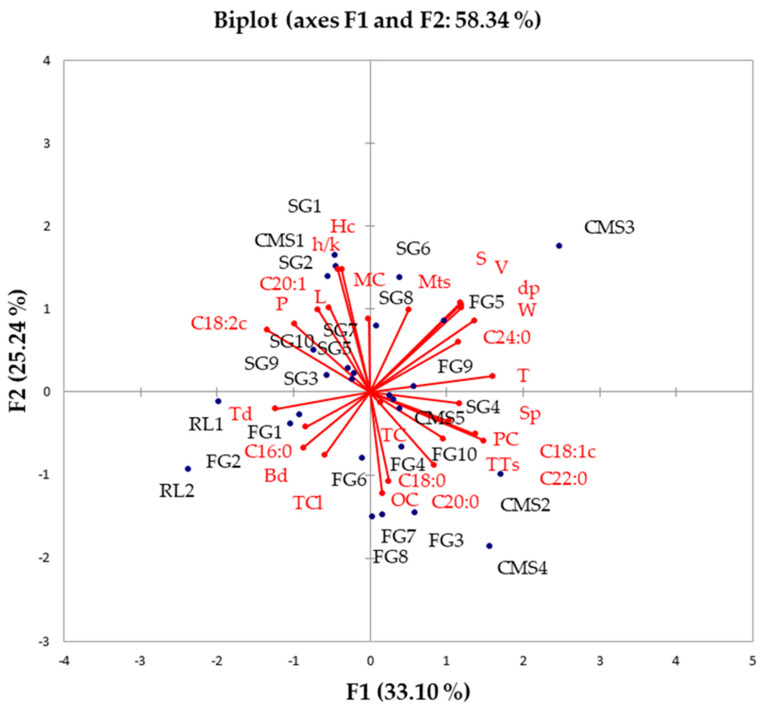
PCA biplot of the nutritional composition and physical properties of sunflowers seeds: Comparison of examined sunflower hybrids and ancestors (L—length, W—width, T—thickness, dp—equivalent diameter, S—surface area, V—seed volume, Sp—sphericity, Hc—hull content, h/k—hull/kernel ratio, Mts—1000-seed mass, Td—true density, Bd—bulk density, P—porosity, MC—moisture content, OC—oil content, PC—protein content, TTs—total tocopherol content in seed, MCc—moisture content in cake, OCs—oil content in cake, PCc—protein content in cake, TPc—polyphenol content in cake, TC—carotenoid content in oil, TCl—chlorophyll content in oil).

**Figure 3 foods-13-01882-f003:**
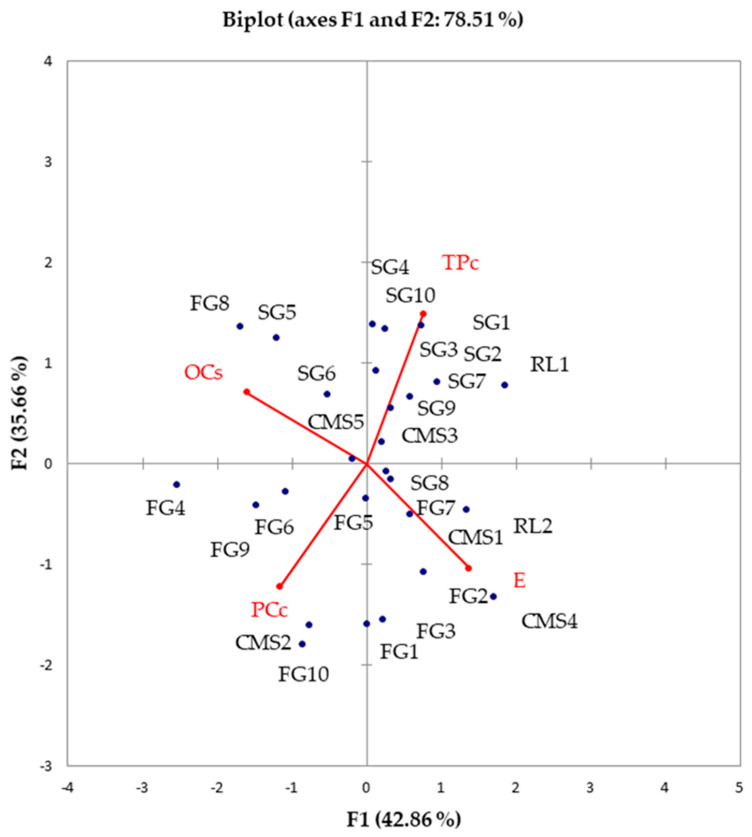
PCA biplot: nutritional composition, physical properties of sunflower seeds, and processing parameters of cake of examined sunflower hybrids and ancestors (OCs—oil content in cake, PCc—protein content in cake, TPc—polyphenol content in cake, E—oil extraction efficiency).

**Table 1 foods-13-01882-t001:** The basic chemical composition of sunflower seed cytoplasmic male sterile paternal lines (CMS1–CMS5), restorer lines (RL1 and RL2), experimental hybrids of the F1 generation (FG1–FG10), as well as the F2 generation (SG1–SG10).

Sample	MC (%)	OC (%)	PC (%)	TTs (mg kg^−1^)
CMS1	6.73 ± 0.26 ^i^	22.53 ± 0.27 ^bc^	19.05 ± 0.02 ^h^	63.95 ± 2.12 ^a^
CMS2	6.02 ± 0.08 ^de^	30.74 ± 0.05 ^i^	20.58 ± 0.32 ^jk^	131.4 ± 18.62 ^bcdefg^
CMS3	7.04 ± 0.18 ^j^	23.29 ± 0.21 ^bcd^	19.86 ± 0.03 ^i^	186.29 ± 21.11 ^h^
CMS4	5.66 ± 0.04 ^b^	35.82 ± 0.39 ^j^	18.52 ± 0.07 ^fgh^	254.51 ± 28.23 ^i^
CMS5	5.97 ± 0.05 ^cd^	28.89 ± 0.38 ^hi^	22.70 ± 0.37 ^m^	185.91 ± 16.12 ^h^
RL1	6.73 ± 0.01 ^i^	24.23 ± 0.36 ^bcde^	14.35 ± 0.09 ^c^	122.01 ± 11.02 ^bcdef^
RL2	6.34 ± 0.05 ^fgh^	31.13 ± 0.82 ^i^	13.13 ± 0.10 ^a^	98.1 ± 5.26 ^abc^
FG1	6.09 ± 0.04 ^de^	25.76 ± 0.74 ^efg^	17.90 ± 0.07 ^ef^	95.24 ± 6.25 ^ab^
FG2	5.72 ± 0.07 ^b^	26.06 ± 1.10 ^efg^	18.01 ± 0.09 ^ef^	101.21 ± 6.94 ^abc^
FG3	5.73 ± 0.06 ^b^	27.48 ± 1.28 ^gh^	20.37 ± 0.33 ^ijk^	126.85 ± 28.15 ^bcdef^
FG4	6.08 ± 0.04 ^de^	23.92 ± 0.12 ^bcde^	20.12 ± 0.06 ^ij^	131.23 ± 9.26 ^bcdefg^
FG5	6.38 ± 0.01 ^gh^	19.35 ± 1.24 ^a^	19.77 ± 0.43 ^i^	112.14 ± 3.20 ^bcd^
FG6	6.09 ± 0.04 ^de^	25.09 ± 0.78 ^def^	21.71 ± 0.35 ^l^	156.04 ± 6.72 ^defgh^
FG7	5.41 ± 0.04 ^a^	28.84 ± 0.49 ^hi^	16.72 ± 0.03 ^d^	173.12 ± 18.17 ^gh^
FG8	5.65 ± 0.04 ^ab^	27.15 ± 0.25 ^fgh^	16.45 ± 0.02 ^d^	141.72 ± 11.64 ^cdefgh^
FG9	5.66 ± 0.04 ^b^	24.10 ± 0.30 ^bcde^	22.03 ± 0.10 ^l^	159.65 ± 14.06 ^efgh^
FG10	5.55 ± 0.07 ^ab^	23.78 ± 0.61 ^bcde^	20.97 ± 0.20 ^k^	123.57 ± 9.58 ^bcdef^
SG1	5.75 ± 0.01 ^bc^	23.83 ± 0.38 ^bcde^	13.43 ± 0.34 ^ab^	126.20 ± 10.55 ^bcdef^
SG2	5.66 ± 0.07 ^b^	22.45 ± 1.60 ^bc^	13.89 ± 0.29 ^bc^	102.32 ± 9.48 ^abc^
SG3	6.14 ± 0.02 ^defg^	23.16 ± 0.95 ^bcd^	16.62 ± 0.35 ^d^	130.43 ± 14.22 ^bcdefg^
SG4	6.12 ± 0.01 ^def^	23.86 ± 0.78 ^bcde^	18.85 ± 0.06 ^gh^	123.34 ± 16.05 ^bcdef^
SG5	6.17 ± 0.04 ^defg^	24.38 ± 0.29 ^cde^	17.63 ± 0.15 ^e^	115.23 ± 7.05 ^bcde^
SG6	6.57 ± 0.04 ^hi^	23.12 ± 0.20 ^bcd^	16.55 ± 0.05 ^d^	136.09 ± 8.27 ^bcdefg^
SG7	6.24 ± 0.10 ^efg^	26.90 ± 0.88 ^fgh^	16.36 ± 0.06 ^d^	141.71 ± 18.08 ^cdefgh^
SG8	6.15 ± 0.04 ^defg^	25.63 ± 0.27 ^efg^	16.46 ± 0.10 ^d^	140.62 ± 18.69 ^bcdefgh^
SG9	6.01 ± 0.03 ^de^	25.20 ± 1.29 ^defg^	18.21 ± 0.09 ^efg^	162.28 ± 20.73 ^fgh^
SG10	6.03 ± 0.02 ^de^	21.98 ± 0.14 ^b^	16.27 ± 0.03 ^d^	114.88 ± 5.84 ^bcde^

MC—moisture content, OC—oil content, PC—protein content, and TTs—total tocopherol content in seeds. Average value ± SD, n = 3. Values with the same letter in a column are not significantly different at 5% according to Tukey’s test.

**Table 2 foods-13-01882-t002:** Dimensional and geometric properties of sunflower seed cytoplasmic male sterile paternal lines (CMS1–CMS5), restorer lines (RL1 and RL2), and hybrids of the F1 generation (FG1–FG10) and F2 generation (SG1–SG10).

Sample	L (mm)	W (mm)	T (mm)	dp (mm)	S (mm^2^)	V (mm^3^)	Sp
CMS1	19.23 ± 1.02 ^m^	6.29 ± 0.55 ^bcde^	3.98 ± 0.46 ^bcde^	7.82 ± 0.55 ^fgh^	192.91 ± 26.99 ^efg^	253.88 ± 52.47 ^efgh^	0.41 ± 0.02 ^a^
CMS2	12.95 ± 0.27 ^abcde^	7.39 ± 0.26 ^ef^	5.60 ± 0.57 ^gh^	8.11 ± 0.37 ^ghi^	206.90 ± 18.78 ^fgh^	280.77 ± 38.18 ^gh^	0.63 ± 0.03 ^ab^
CMS3	12.58 ± 0.96 ^abcd^	9.04 ± 0.82 ^g^	6.12 ± 0.61 ^h^	8.84 ± 0.62 ^i^	246.80 ± 34.68 ^h^	367.37 ± 77.03 ^i^	0.70 ± 0.04 ^b^
CMS4	12.80 ± 0.39 ^abcd^	6.76 ± 0.45 ^cde^	4.74 ± 0.31 ^defg^	7.43 ± 0.36 ^defgh^	173.57 ± 16.76 ^defg^	215.82 ± 31.50 ^cdefgh^	0.58 ± 0.02 ^ab^
CMS5	15.68 ± 0.73 ^hijk^	6.72 ± 0.51 ^bcde^	4.63 ± 0.30 ^defg^	7.86 ± 0.32 ^fgh^	194.35 ± 16.07 ^efg^	255.48 ± 31.74 ^fgh^	0.50 ± 0.03 ^ab^
RL1	13.88 ± 1.09 ^cdefg^	4.66 ± 0.46 ^a^	3.13 ± 0.34 ^ab^	5.86 ± 0.48 ^ab^	108.54 ± 17.41 ^ab^	107.40 ± 25.30 ^ab^	0.42 ± 0.03 ^a^
RL2	14.08 ± 1.00 ^defgh^	4.53 ± 0.46 ^a^	2.89 ± 0.41 ^a^	5.67 ± 0.45 ^a^	101.52 ± 15.84 ^a^	97.09 ± 22.12 ^a^	0.40 ± 0.03 ^a^
FG1	17.18 ± 0.90 ^k^	5.55 ± 0.47 ^ab^	3.35 ± 0.51 ^abc^	6.80 ± 0.40 ^cde^	145.83 ± 17.46 ^bcd^	166.51 ± 30.43 ^abcd^	0.40 ± 0.02 ^a^
FG2	17.17 ± 2.03 ^k^	5.67 ± 0.64 ^abc^	3.33 ± 0.30 ^abc^	6.84 ± 0.50 ^cde^	147.80 ± 21.89 ^bcd^	170.37 ± 38.40 ^abcde^	0.40 ± 0.05 ^a^
FG3	12.50 ± 0.73 ^abcd^	6.36 ± 0.56 ^bcde^	4.57 ± 0.67 ^def^	7.12 ± 0.58 ^cdef^	160.12 ± 25.86 ^cde^	192.43 ± 46.41 ^bcdef^	0.57 ± 0.04 ^ab^
FG4	13.12 ± 0.75 ^bcdef^	6.53 ± 0.87 ^bcde^	4.59 ± 0.69 ^defg^	7.29 ± 0.56 ^cdefg^	167.85 ± 25.54 ^cdef^	206.30 ± 46.82 ^cdefg^	0.56 ± 0.05 ^ab^
FG5	11.89 ± 0.52 ^ab^	8.02 ± 1.06 ^fg^	5.36 ± 0.80 ^fgh^	7.95 ± 0.56 ^fghi^	199.55 ± 28.86 ^efg^	267.17 ± 59.38 ^fgh^	0.67 ± 0.06 ^ab^
FG6	11.33 ± 0.91 ^a^	6.68 ± 0.85 ^bcde^	4.11 ± 0.86 ^bcde^	6.75 ± 0.76 ^bcd^	144.70 ± 31.77 ^abcd^	166.70 ± 53.64 ^abcd^	0.60 ± 0.06 ^ab^
FG7	12.25 ± 0.52 ^abc^	5.71 ± 0.56 ^abc^	3.77 ± 0.47 ^abcd^	6.39 ± 0.44 ^abc^	129.01 ± 18.20 ^abc^	138.84 ± 29.83 ^abc^	0.52 ± 0.02 ^ab^
FG8	12.37 ± 0.27 ^abc^	5.86 ± 0.61 ^bcd^	4.35 ± 0.74 ^cdef^	6.78 ± 0.52 ^bcde^	145.09 ± 22.01 ^abcd^	165.80 ± 37.44 ^abcd^	0.55 ± 0.04 ^ab^
FG9	15.94 ± 0.86 ^ijk^	6.92 ± 0.56 ^def^	4.63 ± 0.57 ^defg^	7.97 ± 0.40 ^fghi^	199.95 ± 19.95 ^efg^	266.92 ± 39.97 ^fgh^	0.50 ± 0.02 ^ab^
FG10	15.53 ± 1.01 ^ghijk^	7.00 ± 0.67 ^def^	4.82 ± 0.53 ^efg^	8.04 ± 0.54 ^fghi^	204.02 ± 28.63 ^efgh^	276.06 ± 60.29 ^fgh^	0.52 ± 0.03 ^ab^
SG1	20.68 ± 1.29 ^m^	7.01 ± 0.62 ^def^	3.91 ± 0.60 ^abcde^	8.25 ± 0.60 ^hi^	214.63 ± 31.80 ^gh^	298.15 ± 67.00 ^hi^	0.40 ± 0.02 ^a^
SG2	18.89 ± 1.03 ^l^	6.70 ± 0.69 ^bcde^	3.91 ± 0.35 ^abcde^	7.89 ± 0.43 ^fgh^	196.09 ± 21.55 ^efg^	259.43 ± 42.65 ^fgh^	0.42 ± 0.03 ^a^
SG3	15.23 ± 1.12 ^ghij^	6.36 ± 1.01 ^bcde^	3.91 ± 0.82 ^abcde^	7.19 ± 0.87 ^cdefg^	164.72 ± 38.09 ^cdef^	202.91 ± 67.36 ^cdefg^	0.47 ± 0.04 ^ab^
SG4	14.49 ± 1.21 ^efghi^	7.17 ± 0.65 ^ef^	4.44 ± 0.42 ^def^	7.70 ± 0.28 ^efgh^	186.41 ± 13.45 ^defg^	239.85 ± 26.09 ^defgh^	0.54 ± 0.05 ^ab^
SG5	13.36 ± 1.25 ^bcdef^	6.80 ± 0.55 ^cde^	4.11 ± 0.59 ^bcde^	7.18 ± 0.65 ^cdefg^	163.32 ± 29.19 ^cdef^	198.68 ± 52.04 ^cdefg^	0.54 ± 0.04 ^ab^
SG6	14.79 ± 0.43 ^fghij^	7.99 ± 1.07 ^fg^	4.39 ± 1.00 ^def^	8.00 ± 0.93 ^fghi^	203.51 ± 48.70 ^efgh^	278.77 ± 102.43 ^gh^	0.54 ± 0.06 ^ab^
SG7	15.62 ± 0.84 ^hijk^	7.16 ± 1.10 ^ef^	4.01 ± 0.75 ^bcde^	7.58 ± 0.54 ^defgh^	181.22 ± 25.16 ^defg^	231.13 ± 46.87 ^defgh^	0.49 ± 0.04 ^ab^
SG8	16.15 ± 0.83 ^ijk^	7.24 ± 0.23 ^ef^	4.24 ± 0.35 ^cde^	7.90 ± 0.30 ^fgh^	196.27 ± 14.70 ^efg^	259.16 ± 29.17 ^fgh^	0.49 ± 0.02 ^ab^
SG9	16.38 ± 0.93 ^jk^	6.62 ± 0.63 ^bcde^	4.00 ± 0.69 ^bcde^	7.53 ± 0.46 ^defgh^	178.54 ± 22.30 ^defg^	225.68 ± 43.09 ^defgh^	0.46 ± 0.04 ^ab^
SG10	16.34 ± 1.33 ^jk^	6.57 ± 0.41 ^bcde^	3.81 ± 0.36 ^abcde^	7.36 ± 0.32 ^cdefgh^	168.64 ± 15.42 ^cdef^	207.80 ± 27.11 ^cdefg^	0.72 ± 0.80 ^b^

L—length, W—width, T—thickness, dp—equivalent diameter, S—surface area, V—seed volume, Sp—sphericity. Average value ± SD, n = 3. Values with the same letter in a column are not significantly different at 5% according to Tukey’s test.

**Table 3 foods-13-01882-t003:** Technical and technological characteristics of sunflower seed cytoplasmic male sterile paternal lines (CMS1–CMS5), restorer lines (RL1 and RL2), and hybrids of the F1 generation (FG1–FG10) and F2 generation (SG1–SG10).

Sample	Hc (%)	h/k	Mts (g)	Bd (kg m^−3^)	Td (kg m^−3^)	P (%)
CMS1	47.88 ± 0.97 ^n^	0.92 ± 0.04 ^l^	110.78 ± 2.68 ^ghi^	294.00 ± 2.80 ^c^	561.93 ± 4.83 ^de^	47.67 ± 0.95 ^lmn^
CMS2	34.89 ± 0.86 ^cdef^	0.54 ± 0.02 ^cdef^	109.01 ± 0.98 ^gh^	311.80 ± 0.60 ^de^	518.88 ± 2.28 ^bc^	39.91 ± 0.15 ^b^
CMS3	46.03 ± 0.17 ^mn^	0.86 ± 0.01 ^kl^	116.58 ± 1.82 ^ijk^	268.40 ± 3.20 ^a^	460.44 ± 2.71 ^a^	41.71 ± 0.35 ^cd^
CMS4	29.36 ± 0.59 ^a^	0.42 ± 0.01 ^ab^	79.20 ± 1.20 ^c^	317.40 ± 0.20 ^ef^	531.60 ± 1.46 ^c^	40.29 ± 0.20 ^bc^
CMS5	37.59 ± 2.95 ^fgh^	0.60 ± 0.08 ^defg^	131.52 ± 1.38 ^mn^	376.60 ± 0.60 ^p^	608.09 ± 5.37 ^h^	38.07 ± 0.45 ^a^
RL1	43.58 ± 0.12 ^klm^	0.79 ± 0.03 ^ijk^	53.09 ± 0.63 ^a^	344.00 ± 0.40 ^hij^	631.60 ± 3.91 ^i^	45.53 ± 0.40 ^hij^
RL2	37.67 ± 1.75 ^fgh^	0.61 ± 0.05 ^defg^	53.62 ± 0.84 ^a^	366.60 ± 0.60 ^no^	711.86 ± 5.28 ^m^	48.50 ± 0.47 ^n^
FG1	40.42 ± 0.44 ^hijk^	0.68 ± 0.01 ^ghi^	90.10 ± 1.33 ^ef^	349.20 ± 3.20 ^jk^	650.06 ± 3.16 ^j^	46.28 ± 0.23 ^jklm^
FG2	40.31 ± 0.64 ^hijk^	0.68 ± 0.02 ^ghi^	105.27 ± 0.81 ^g^	373.80 ± 1.40 ^op^	673.65 ± 2.15 ^k^	44.51 ± 0.38 ^ghi^
FG3	31.74 ± 0.35 ^abc^	0.47 ± 0.01 ^abc^	86.20 ± 1.22 ^cde^	349.80 ± 0.60 ^jk^	603.19 ± 1.76 ^gh^	42.01 ± 0.07 ^cde^
FG4	34.20 ± 0.54 ^bcde^	0.52 ± 0.01 ^bcde^	89.11 ± 1.34 ^de^	339.60 ± 0.80 ^ghi^	589.46 ± 0.36 ^fg^	42.39 ± 0.17 ^def^
FG5	39.48 ± 0.07 ^ghij^	0.65 ± 0.00 ^fgh^	96.46 ± 0.35 ^f^	284.00 ± 2.00 ^b^	509.11 ± 12.43 ^b^	44.20 ± 0.97 ^gh^
FG6	37.02 ± 0.06 ^efg^	0.59 ± 0.00 ^defg^	82.23 ± 0.71 ^cd^	353.20 ± 0.01 ^kl^	587.38 ± 6.06 ^f^	39.86 ± 0.62 ^b^
FG7	33.12 ± 0.93 ^bcd^	0.50 ± 0.02 ^abcd^	69.10 ± 3.53 ^b^	337.20 ± 6.40 ^gh^	565.37 ± 1.81 ^de^	40.36 ± 1.32 ^bc^
FG8	31.29 ± 1.56 ^ab^	0.39 ± 0.12 ^a^	69.19 ± 0.93 ^b^	334.40 ± 0.40 ^g^	576.13 ± 8.56 ^ef^	41.95 ± 0.79 ^cde^
FG9	35.02 ± 0.89 ^def^	0.54 ± 0.02 ^cdef^	127.40 ± 3.17 ^lm^	360.60 ± 2.60 ^mn^	649.37 ± 2.27 ^j^	44.47 ± 0.59 ^ghi^
FG10	31.54 ± 0.33 ^ab^	0.46 ± 0.01 ^abc^	137.11 ± 0.79 ^no^	361.00 ± 0.20 ^mn^	670.94 ± 7.49 ^k^	46.19 ± 0.63 ^ijkl^
SG1	46.74 ± 0.99 ^mn^	0.88 ± 0.03 ^kl^	123.21 ± 5.43 ^kl^	305.40 ± 0.20 ^d^	587.04 ± 0.97 ^f^	47.98 ± 0.12 ^mn^
SG2	45.10 ± 2.20 ^lmn^	0.82 ± 0.07 ^jkl^	121.73 ± 1.37 ^kl^	307.80 ± 0.20 ^e^	590.49 ± 5.12 ^fg^	47.87 ± 0.49 ^lmn^
SG3	41.07 ± 0.19 ^ijk^	0.70 ± 0.01 ^ghi^	116.55 ± 4.32 ^ijk^	365.60 ± 2.40 ^n^	648.00 ± 3.64 ^j^	43.58 ± 0.05 ^efg^
SG4	39.06 ± 1.28 ^ghij^	0.64 ± 0.03 ^fgh^	113.53 ± 0.30 ^hi^	346.80 ± 2.00 ^ijk^	631.06 ± 1.54 ^i^	45.04 ± 0.45 ^ghij^
SG5	37.88 ± 0.82 ^fghi^	0.61 ± 0.02 ^defgh^	89.00 ± 1.89 ^de^	338.80 ± 0.80 ^gh^	607.76 ± 0.54 ^h^	44.26 ± 0.18 ^gh^
SG6	45.38 ± 0.15 ^mn^	0.83 ± 0.01 ^jkl^	114.22 ± 3.60 ^hij^	322.60 ± 3.40 ^f^	559.63 ± 3.13 ^d^	42.36 ± 0.29 ^def^
SG7	38.03 ± 0.05 ^fghi^	0.61 ± 0.00 ^efgh^	113.52 ± 2.61 ^hi^	358.00 ± 0.80 ^lm^	666.91 ± 8.00 ^k^	46.32 ± 0.52 ^jklm^
SG8	41.91 ± 0.87 ^jkl^	0.72 ± 0.03 ^hij^	125.56 ± 3.66 ^lm^	340.00 ± 4.80 ^ghi^	645.59 ± 1.19 ^ij^	47.33 ± 0.84 ^klmn^
SG9	36.98 ± 0.35 ^efg^	0.59 ± 0.01 ^defg^	139.63 ± 6.49 ^o^	394.80 ± 2.80 ^r^	704.93 ± 1.31 ^lm^	44.00 ± 0.29 ^fgh^
SG10	39.81 ± 0.67 ^ghij^	0.66 ± 0.02 ^gh^	120.95 ± 3.80 ^jkl^	377.20 ± 1.20 ^p^	693.59 ± 5.25 ^l^	45.61 ± 0.58 ^hijk^

Hc—hull content, h/k—hull/kernel ratio, Mts—1000-seed mass, Td—true density, Bd—bulk density, P—porosity. Average value ± SD, n = 3. Values with the same letter in a column are not significantly different at 5% according to Tukey’s test.

**Table 4 foods-13-01882-t004:** Comprehensive analysis of moisture, fats, protein, and polyphenol contents in cake, and carotenoids and chlorophyll (pheophytin A) in oil, across different genotypes.

Sample	Cake				Oil	
	MCc (%)	OCs (%)	PCc (%)	TPc (mg CGA eq kg^−1^)	TC (mg β-C eq kg^−1^)	TCl (mg Phe A kg^−1^)
CMS1	5.33 ± 0.01 ^b^	17.04 ± 0.05 ^a^	23.55 ± 0.08 ^l^	691.29 ± 3.35 ^defgh^	5.25 ± 0.01 ^abcd^	0.37 ± 0.02 ^cde^
CMS2	6.17 ± 0.15 ^n^	22.52 ± 0.01 ^hij^	27.48 ± 0.04 ^o^	602.27 ± 2.51 ^abc^	7.17 ± 0.01 ^def^	0.48 ± 0.01 ^defg^
CMS3	5.64 ± 0.03 ^d^	32.16 ± 1.80 ^l^	22.22 ± 0.11 ^ijk^	723.90 ± 5.02 ^ghijkl^	14.19 ± 2.90 ^jk^	0.00 ± 0.00 ^a^
CMS4	5.82 ± 0.04 ^efg^	44.38 ± 1.86 ^m^	20.32 ± 0.24 ^ef^	625.73 ± 12.01 ^cdef^	11.82 ± 0.08 ^ijk^	0.44 ± 0.01 ^def^
CMS5	5.88 ± 0.04 ^fghi^	22.20 ± 0.14 ^ghi^	22.58 ± 0.23 ^k^	692.45 ± 4.99 ^defghi^	11.40 ± 0.01 ^hij^	0.41 ± 0.01 ^def^
RL1	5.85 ± 0.05 ^fg^	18.08 ± 0.06 ^abcd^	17.17 ± 0.06 ^a^	780.66 ± 14.51 ^ijkl^	11.35 ± 0.05 ^hij^	0.65 ± 0.02 ^ghi^
RL2	5.94 ± 0.09 ^fghijklm^	17.26 ± 0.46 ^ab^	18.05 ± 0.10 ^b^	618.40 ± 6.96 ^bcde^	11.97 ± 0.03 ^ijk^	0.88 ± 0.02 ^j^
FG1	5.47 ± 0.06 ^bc^	18.75 ± 1.15 ^abcde^	22.44 ± 0.39 ^jk^	516.26 ± 58.82 ^a^	9.55 ± 0.09 ^fghi^	0.57 ± 0.01 ^fghi^
FG2	5.60 ± 0.01 ^cd^	17.49 ± 0.91 ^ab^	22.75 ± 0.13 ^k^	636.46 ± 28.35 ^cdefg^	11.78 ± 0.19 ^ijk^	0.57 ± 0.28 ^fghi^
FG3	6.02 ± 0.00 ^hijklmn^	18.49 ± 0.79 ^abcd^	24.54 ± 0.10 ^m^	635.48 ± 58.60 ^cdefg^	10.24 ± 0.18 ^ghi^	0.70 ± 0.03 ^i^
FG4	6.04 ± 0.02 ^ijklmn^	24.82 ± 1.20 ^jk^	24.94 ± 0.08 ^m^	534.71 ± 60.48 ^ab^	8.10 ± 0.02 ^efg^	0.21 ± 0.01 ^bc^
FG5	5.68 ± 0.02 ^de^	17.61 ± 0.94 ^abc^	21.81 ± 0.13 ^hi^	610.02 ± 39.78 ^bcd^	13.39 ± 0.40 ^jk^	0.14 ± 0.01 ^ab^
FG6	5.81 ± 0.04 ^ef^	21.91 ± 0.62 ^ghi^	25.96 ± 0.12 ^n^	698.01 ± 35.81 ^defghijk^	13.83 ± 0.08 ^k^	0.38 ± 0.01 ^cde^
FG7	5.91 ± 0.02 ^fghijk^	20.47 ± 0.84 ^defgh^	21.97 ± 0.02 ^hij^	702.37 ± 29.77 ^efghijk^	5.87 ± 0.10 ^cde^	0.00 ± 0.00 ^a^
FG8	6.07 ± 0.01 ^lmn^	26.30 ± 0.08 ^k^	21.20 ± 0.26 ^g^	709.54 ± 17.30 ^fghijk^	9.35 ± 0.07 ^fgh^	1.02 ± 0.03 ^j^
FG9	6.09 ± 0.02 ^mn^	22.21 ± 0.66 ^ghi^	26.53 ± 0.06 ^n^	666.62 ± 26.02 ^cdefgh^	7.54 ± 0.36 ^def^	0.35 ± 0.01 ^cd^
FG10	6.06 ± 0.04 ^klmn^	19.45 ± 0.83 ^bcdef^	25.98 ± 0.24 ^n^	516.12 ± 62.55 ^a^	7.25 ± 2.63 ^cde^	0.67 ± 0.01 ^hi^
SG1	5.15 ± 0.05 ^a^	20.00 ± 0.90 ^cdefg^	17.45 ± 0.05 ^a^	806.66 ± 5.34 ^lm^	11.57 ± 0.27 ^hijk^	0.54 ± 0.01 ^efghi^
SG2	5.10 ± 0.07 ^a^	18.47 ± 0.10 ^abcd^	17.05 ± 0.32 ^a^	740.76 ± 5.32 ^hijkl^	10.29 ± 0.14 ^ghi^	0.00 ± 0.00 ^a^
SG3	5.98 ± 0.00 ^ghijklm^	20.26 ± 0.09 ^defgh^	20.43 ± 0.24 ^f^	785.72 ± 1.18 ^klm^	3.06 ± 0.04 ^a^	0.00 ± 0.00 ^a^
SG4	5.92 ± 0.04 ^fghijkl^	20.96 ± 0.02 ^efgh^	21.56 ± 0.06 ^gh^	872.36 ± 6.25 ^m^	3.37 ± 0.05 ^ab^	0.50 ± 0.02 ^defgh^
SG5	5.69 ± 0.02 ^de^	24.14 ± 0.23 ^ijk^	19.63 ± 0.03 ^d^	695.79 ± 1.19 ^defghijk^	6.31 ± 0.21 ^cde^	0.41 ± 0.02 ^def^
SG6	5.88 ± 0.04 ^fgh^	21.65 ± 0.45 ^fgh^	19.05 ± 0.20 ^c^	663.61 ± 4.45 ^cdefgh^	6.48 ± 0.33 ^cde^	0.37 ± 0.01 ^cde^
SG7	5.89 ± 0.11 ^fghij^	20.40 ± 0.21 ^defgh^	19.72 ± 0.05 ^d^	782.87 ± 3.34 ^jkl^	4.08 ± 0.13 ^abc^	0.06 ± 0.01 ^ab^
SG8	6.04 ± 0.01 ^hijklmn^	20.11 ± 0.01 ^defg^	18.92 ± 0.29 ^c^	639.61 ± 13.64 ^cdefg^	5.78 ± 0.08 ^bcde^	0.00 ± 0.00 ^a^
SG9	6.10 ± 0.01 ^mn^	20.31 ± 0.13 ^defgh^	19.79 ± 0.18 ^de^	744.73 ± 8.34 ^hijkl^	4.42 ± 0.12 ^abc^	0.42 ± 0.01 ^def^
SG10	6.04 ± 0.01 ^jklmn^	20.06 ± 0.02 ^defg^	19.33 ± 0.25 ^cd^	810.83 ± 5.47 ^lm^	5.56 ± 0.16 ^bcd^	0.13 ± 0.00 ^ab^

MCc—moisture content in cake, OCs—oil content in cake, PCc—protein content in cake, TPc—polyphenol content in cake, TC—carotenoid content in oil, TCl—chlorophyll content in oil. Average value ± SD, n = 3. Values with the same letter in a column are not significantly different at 5% according to Tukey’s test.

**Table 5 foods-13-01882-t005:** Pearson’s correlation coefficients among the analyzed nutritional composition (seed, oil, and cake), physical properties of sunflower seeds, and processing parameter variables.

	Mts	dp	S	V	Sp	h/k	Td	P	OC	PC	TTs	C16:0	C18:0	C18:1	C18:2	C20:0	C20:1	C22:0	C24:0	TC	TCl	E	Ocs	PCc	TPc
Mts	**1**	**0.80**	**0.78**	**0.76**	0.04	0.25	0.14	0.14	**−0.38**	0.28	0.02	−0.22	−0.30	0.09	−0.01	−0.13	0.12	−0.01	0.23	−0.37	−0.37	−0.23	−0.12	0.12	0.08
dp		**1**	**0.99**	**0.99**	0.34	0.28	**−0.43**	−0.05	−0.32	0.37	0.18	**−0.47**	−0.20	0.37	−0.29	0.10	0.8	**0.40**	**0.60**	−0.10	**−0.43**	−0.20	0.22	0.20	0.01
S			**1**	**0.99**	0.34	0.31	**−0.45**	−0.04	−0.32	0.35	0.18	**−0.45**	−0.21	0.36	−0.28	0.09	0.10	**0.40**	**0.63**	−0.07	**−0.43**	−0.19	0.22	0.19	0.02
V				**1**	0.34	0.34	**−0.46**	−0.03	−0.32	0.33	0.18	**−0.44**	−0.23	0.35	−0.27	0.07	0.13	**0.39**	**0.65**	−0.04	**−0.42**	−0.18	0.22	0.17	0.03
Sp					**1**	−0.28	**−0.48**	**−0.53**	−0.08	**0.44**	0.38	−0.26	−0.05	**0.54**	**−0.50**	0.18	−0.24	**0.51**	0.28	0.11	−0.32	−0.19	**0.42**	0.32	0.00
h/k						**1**	−0.14	**0.48**	**−0.55**	**−0.41**	**−0.42**	−0.13	**−0.45**	**−0.47**	**0.56**	**−0.44**	**0.54**	**−0.43**	0.27	0.05	−0.28	−0.11	−0.31	**−0.46**	**0.41**
Td							**1**	**0.49**	−0.02	−0.26	−0.32	**0.44**	−0.07	**−0.46**	**0.45**	−0.34	0.11	**−0.66**	**−0.65**	**−0.39**	0.26	−0.05	**−0.48**	−0.21	0.02
P								**1**	**−0.40**	**−0.60**	**−0.66**	0.09	−0.13	**−0.56**	**0.57**	−0.34	**0.43**	**−0.60**	−0.20	−0.20	0.03	−0.12	**−0.50**	**−0.48**	0.08
OC									**1**	0.02	**0.54**	0.14	**0.56**	0.21	−0.34	**0.52**	−0.34	**0.43**	−0.08	0.16	0.34	**0.70**	**0.50**	0.09	−0.18
PC										**1**	0.30	0.10	0.06	**0.52**	**−0.54**	0.29	**−0.40**	**0.53**	0.31	0.08	−0.02	−0.06	0.20	**0.86**	−0.36
TTs											**1**	−0.16	0.00	**0.54**	**−0.52**	0.20	−0.24	**0.58**	0.37	0.19	−0.20	0.31	**0.81**	0.07	0.07
C16:0												**1**	−0.01	**−0.54**	**0.43**	−0.19	−0.06	−0.32	−0.26	0.25	**0.39**	0.16	−0.25	0.11	−0.03
C18:0													**1**	0.22	**−0.43**	**0.91**	**−0.60**	**0.50**	−0.14	−0.12	0.32	**0.46**	0.17	0.36	−0.28
C18:1c														**1**	**−0.97**	**0.46**	−0.31	**0.76**	**0.42**	0.03	−0.15	0.03	**0.55**	**0.50**	−0.28
C18:2c															**1**	**−0.63**	**0.44**	**−0.84**	**−0.38**	−0.03	0.05	−0.14	**−0.57**	**−0.57**	0.34
C20:0																**1**	**−0.61**	**0.76**	0.15	−0.12	0.05	**0.41**	0.37	**0.50**	−0.30
C20:1																	**1**	**−0.46**	0.00	0.15	−0.08	−0.18	−0.11	**−0.52**	0.06
C22:0																		**1**	**0.57**	0.08	−0.10	0.25	**0.71**	**0.52**	−0.29
C24:0																			**1**	0.15	−0.36	0.08	**0.48**	0.18	0.00
TC																				**1**	0.29	0.28	0.23	0.04	−0.19
TCl																					**1**	0.20	−0.08	0.12	−0.21
E																						**1**	0.23	−0.03	−0.03
Ocs																							**1**	0.05	−0.02
PCc																								**1**	**−0.49**
TPc																									**1**

Values in bold are different from 0 with a significance level of alpha = 0.05.

## Data Availability

The original contributions presented in the study are included in the article, further inquiries can be directed to the corresponding author.

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
