# Peer review of "Assessing Nutritional Characteristics and Bioactive Compound Distribution in Seeds, Oil, and Cake from Confectionary Sunflowers Cultivated in Serbia"

_foods, 2024, doi:10.3390/foods13121882_

Round 1

Reviewer 1 Report

Comments and Suggestions for Authors

The manuscript describes and compares seeds, oils and cake from confectionary sunflowers cultivated in Serbia in terms of their biochemical composition, characteristics of seeds, fatty acids and bioactive components.

The study was well designed.

Here are the specific comments to the manuscript:

line 100: remove one dot

line 149: please specify the soxhlet aparatures - brand name, model, country

line 267: add "]" after 43.

line 306: discussion is missing, a comparison with results of other researchers

line 310: please replace "commercial" with "technological"

line 330: please add and information in the manuscript that oleic acid and SFA are desired from the technological point of view due to their positive effect on oil stability improvement

line 341: please add more details

figures 4 and 5: please add description of some abbreviations used the plot (Td, Bd, TCl, Sp,...)

Author Response

Replies for

Referee 1

Authors comments

Thank you very much for your consideration and invaluable comments on our recent revision. Your suggestions and feedback on the content helped us to improve the manuscript. We appreciate your effort and the time you took to read the manuscript. We carefully went through all of your comments and revised the manuscript accordingly.

Referee comments and Text Location

Comment

Line 100 in introduction section “remove one dot”

According referee’s opinion we removed one dot.

Line 149 in materials and methods

“please specify the soxhlet aparatures - brand name, model, country”

In response to the referee's request, we have provided the specific details of the Soxhlet apparatus, including the brand name, model, and country of origin.

Line 267 in results and discussion “add "]" after 43”

According referee’s opinion we added "]" after 43.

line 306: in results and discussion “discussion is missing, a comparison with results of other researchers”

According referee’s opinion we added discussion.

line 310 in results and discussion

“please replace "commercial" with "technological”

According referee’s opinion we have replaced „commercial” with “technological”.

line 330 in results and discussion

“please add and information in the manuscript that oleic acid and SFA are desired from the technological point of view due to their positive effect on oil stability improvement”

According to the referee's request, we have included the missing information regarding oleic acid and saturated fatty acids (SFA) in the manuscript. These components are technologically significant due to their beneficial impact on improving oil stability.

line 341 in results and discussion

“please add more details”

According referee’s opinion we have added more details.

in results and discussion “Figures 4 and 5: please add description of some abbreviations used the plot (Td, Bd, TCl, Sp,...)”

According referee’s opinion we have added description of some abbreviations used the plots.

 Sincerely yours,

Sincerely yours,

PhD Nada Grahovac, Senior Research Associate

Corresponding author: Nada Grahovac

Institute of Field and Vegetable Crops,

Maksima Gorkog 30, 21101 Novi Sad, Serbia

Tel: +381 21 4898 419; Fax: +381 21 4898 418

E-mail address: [email protected]   

Resubmission Date

10 June 2024

Reviewer 2 Report

Comments and Suggestions for Authors

Line32, please check whether the global consumption of sunflower oil accounts for 75% of all edible vegetable oils, perhaps the FAO Database has relevant data. As I known, the consumption of soybean oil, palm oil and peanut oil accounts for a very high proportion.

Line 135, there should be a space between the value and the temperature symbol oC.

Line 100, need to delete a period.

Line 211, as shown in Equation 6, the physical quantity E representing the oil extraction efficiency is used in italics, and the letters representing physical quantities in other equations (1-5) are also recommended in italics.

Line 229-230, It is recommended to change the title of Table 1 to The basic chemical composition of sunflower seed cytoplasmic male sterile paternal lines (CMS1-CMS5), restorer lines (RL1 and RL2), experimental hybrids of the F1 generation (FG1-FG10), as well as the F2 generation (SG1-SG10).

Line 247, CMS4 ranked first among all samples with 254.51 mg/kg TTs content.

Line 252, references should be provided.

Line 267, the square brackets for the reference 43 are incomplete.

Table 2, Italics are recommended for letters representing physical quantities in table 2, as well as the notes.

Line 297, 53,62 should be 53.62.

Line 279, as the table 2 shows, the parental line CMS1 with a length of 19.23.

Line 326, why is C18:1c used to refer to oleic acid? As well as the legend in Figure 3A. In line 342 oleic acid is represented by C18:1 and linoleic acid by C18:2.

Line 372-376, The description that the difference is not significant is not correct, just as the notes of Table 3 described, values with the same letter in a column are not significantly different at 5% according to Tuckys test. The TC values for CMS3 and SG3 is 14.19 and 3.06 respectively, the difference is significant. As well as the TCl content.

Line 383, 26.53 can not be found in Table 3, the highest value for the F1 and F2 genotypes is 26.30%.

Line 486, 13.89% should be 13.43%.

Author Response

Replies for

Referee 2

Authors comments

Thank you very much for your consideration and invaluable comments on our recent revision. Your suggestions and feedback on the content helped us to improve the manuscript. We appreciate your effort and the time you took to read the manuscript. We carefully went through all of your comments and revised the manuscript accordingly.

Referee comments and Text Location

Comment

Line 32 in introduction section “please check whether the global consumption of sunflower oil accounts for 75% of all edible vegetable oils, perhaps the FAO Database has relevant data. As I known, the consumption of soybean oil, palm oil and peanut oil accounts for a very high proportion.”

According referee’s opinion we have revised the section pertaining to the global consumption of sunflower oil.

Line 135 in materials and methods

“there should be a space between the value and the temperature symbol ºC.”

According referee’s opinion we have inserted space between the value and the temperature symbol ºC.

Line 100 in introduction section “need to delete a period.”

According referee’s opinion we delated a period.

line 211: in results and discussion “as shown in Equation 6, the physical quantity E representing the oil extraction efficiency is used in italics, and the letters representing physical quantities in other equations (1-5) are also recommended in italics.”

By reviewing available manuscripts that addressed the same physical quantities, we noticed that these quantities were not listed in italics. In our case, the software we used to prepare equations (1-6) formatted them in italics and by mistake we left equation 6 in italics.  We have now corrected this error in equation 6 which is consistent with manuscripts that examined the same physical quantity.

line 229-230 in results and discussion

It is recommended to change the title of Table 1 to “The basic chemical composition of sunflower seed cytoplasmic male sterile paternal lines (CMS1-CMS5), restorer lines (RL1 and RL2), experimental hybrids of the F1 generation (FG1-FG10), as well as the F2 generation (SG1-SG10)”.

According referee’s opinion we have corrected title of Table 1.

line 247 in results and discussion

“Line 247, CMS4 ranked first among all samples with 254.51 mg/kg TTs content.”

According to the referee's request, we have added information that was missing in line 237, which refers to the commenting of parental lines with the sentence "Additionally, hybrid CMS4 not only has a high oil content but also boasts the highest total tocopherol content at 254.51 mg/kg in the seed", and not in the part that refers to hybrid combinations (line 247)

line 252 in results and discussion

“references should be provided”

According referee’s opinion we have added references.

Line 267 in results and discussion “, the square brackets for the reference 43 are incomplete.”

According referee’s opinion we have added square brackets for the reference 43.

Line 290 in results and discussion “Table 2, Italics are recommended for letters representing physical quantities in table 2, as well as the notes.”

By reviewing available manuscripts that addressed the same physical quantities, we noticed that these quantities were not listed in italics. In our case, the software we used to prepare equations (1-6) formatted them in italics and by mistake we left equation 6 in italics.  We have now corrected this error in equation 6 which is consistent with manuscripts that examined the same physical quantity. Therefore, we have left the physical quantities as they were stated.

Line 297 in results and discussion “53,62 should be 53.62.”

According referee’s opinion we have corrected 53,62 to 53.62

Line 279 in results and discussion “as the table 2 shows, the parental line CMS1 with a length of 19.23”

According referee’s opinion we have corrected 15.68 to 19.23 mm for parental lines.

Line 326 in results and discussion “why is C18:1c used to refer to oleic acid? As well as the legend in Figure 3A. In line 342 oleic acid is represented by C18:1 and linoleic acid by C18:2”

According referee’s opinion we have corrected C18:1c to C18:1 and C18:2c to C18:2.

Line 372-376 in results and discussion “The description that the difference is not significant is not correct, just as the notes of Table 3 described, values with the same letter in a column are not significantly different at 5% according to Tucky’s test. The TC values for CMS3 and SG3 is 14.19 and 3.06 respectively, the difference is significant. As well as the TCl content.”

According referee’s opinion we have corrected and added “The total carotenoid content (TC) ranged from 3.06 to 14.19 mg/kg in the examined oil samples. The highest content of all examined parental lines possessed mother line CMS3 (14.19 mg kg-1) that was statistically higher from lines and F1 and F2 generations (FG1, FG3, FG4, FG7-FG10 and SG2-SG10). In the case of total chlorophyll content (TCl) it was noticed opposite trend. Namely, this parental line possessed the statistically lowest content (TCl) (0.00 mg kg-1) of all parental lines. Also, it was statistically low-er from lines and F1 and F2 generations (FG1-FG4, FG6 FG8-FG10, SG1, SG4-SG6, SG9 and SG10) , characteristic of unrefined sunflower oil.”

Line 383, in results and discussion

“26.53 can not be found in Table 3, the highest value for the F1 and F2 genotypes is 26.30%.”

According referee’s opinion we have corrected value 26.53 to 26.30%.

Line 486, in results and discussion “13.89% should be 13.43%.”

According referee’s opinion we have corrected value 13.89 to 13.43%.

 Sincerely yours,

Sincerely yours,

PhD Nada Grahovac, Senior Research Associate

Corresponding author: Nada Grahovac

Institute of Field and Vegetable Crops,

Maksima Gorkog 30, 21101 Novi Sad, Serbia

Tel: +381 21 4898 419; Fax: +381 21 4898 418

E-mail address: [email protected]   

Resubmission Date

10 June 2024

Reviewer 3 Report

Comments and Suggestions for Authors

The manuscript entitled “Assessing Nutritional Characteristics and Bioactive Compound Distribution in Sunflower Seeds, Oil and Cake from Confectionary Sunflowers Cultivated in Serbia” investigated the quality of 27 genotypes confectionary sunflower seeds (moisture, oil and protein contents, engineering characteristics of sunflower seeds like seeds dimensions, fatty acid profile and bioactive components of sunflower seed, oil and cake). Results showed the complex dynamics of hybrid seed diversity and their implications for food industry applications. The manuscript is well prepared and the results may be of interest to the journal's readers.

Some minor suggestion for the authors:

- Table 2 should be rearranged (to see the numbers and letters in the columns much better)

- Figures 3A and 3B could be improved (to make them clearer)

- Section 3.3 – compare your results with other studies

Author Response

Replies for

Referee 3

Authors comments

Thank you very much for your consideration and invaluable comments on our recent revision. Your suggestions and feedback on the content helped us to improve the manuscript. We appreciate your effort and the time you took to read the manuscript. We carefully went through all of your comments and revised the manuscript accordingly.

Referee comments and Text Location

Comment

In results and discussion: Table 2 should be rearranged (to see the numbers and letters in the columns much better)

According referee’s opinion we have rearranged Table 2.

In results and discussion: Figures 3A and 3B could be improved (to make them clearer)

According to the referee's opinion we have improved the Figures by increasing the fonts (Fig. 3A) and changing the colors of the columns (Fig. 3B), now it is more visible.

In results and discussion: Section 3.3 – compare your results with other studies

According referee’s opinion we compared results with other studies

 Sincerely yours,

Sincerely yours,

PhD Nada Grahovac, Senior Research Associate

Corresponding author: Nada Grahovac

Institute of Field and Vegetable Crops,

Maksima Gorkog 30, 21101 Novi Sad, Serbia

Tel: +381 21 4898 419; Fax: +381 21 4898 418

E-mail address: [email protected]   

Resubmission Date

10 June 2024

Reviewer 4 Report

Comments and Suggestions for Authors

The paper ”Assessing Nutritional Characteristics and Bioactive Compound Distribution in Sunflower Seeds, Oil and Cake from Confectionary Sunflowers Cultivated in Serbia” investigated the physico-chemical composition in sunflower seeds, oil, and cake from parental lines and their F1 and F2 generations derived through the conventional crossing methods.

The results of the present paper are important given that wild sunflower seeds are recognized as a promising source of valuable phytochemical compounds, such as phenols and polyunsaturated fatty acid.

The experimental design and analysis methodologies used are suitable and feasible. There are sufficient details in order to replicate the experimental procedures and analysis.

The findings of the researchers attest the complex dynamics of hybrid seed diversity and their implications for food industry. Moreover, the study emphases the genetic potential for enhancing the nutritional profile of sunflower seed oils and the influence of various factors on sunflower seed oil and cake composition.

In conclusion, the present research provides valuable information about the selection of genotypes that meet specific nutritional and processing standards, considering their utilization as raw materials within the food industry. Other than that, congratulations to the authors for this clear and well written research.

Author Response

Replies for

Referee 4

Authors comment

We sincerely appreciate your dedicated effort and the time you invested in reviewing the manuscript, along with your positive comments.

Sincerely yours,

PhD Nada Grahovac, Senior Research Associate

Corresponding author: Nada Grahovac

Institute of Field and Vegetable Crops,

Maksima Gorkog 30, 21101 Novi Sad, Serbia

Tel: +381 21 4898 419; Fax: +381 21 4898 418

E-mail address: [email protected]   

Resubmission Date

10 June 2024
